# The Effect of JAK Inhibitor on the Survival, Anagen Re-Entry, and Hair Follicle Immune Privilege Restoration in Human Dermal Papilla Cells

**DOI:** 10.3390/ijms21145137

**Published:** 2020-07-20

**Authors:** Jung Eun Kim, Yu Jin Lee, Hye Ree Park, Dong Geon Lee, Kwan Ho Jeong, Hoon Kang

**Affiliations:** Department of Dermatology, Eunpyeong St. Mary’s Hospital, College of Medicine, The Catholic University of Korea, Seoul 03312, Korea; cindyeujine1@naver.com (Y.J.L.); 5953hari@naver.com (H.R.P.); dlcjdgnl@naver.com (D.G.L.); jaykh86@gmail.com (K.H.J.)

**Keywords:** JAK inhibitor, hair follicle, dermal papilla cell

## Abstract

Topical or systemic administration of JAK inhibitors has been shown to be a new treatment modality for severe alopecia areata (AA). Some patients show a good response to JAK inhibitors, but frequently relapse after cessation of the treatment. There have been no guidelines about the indications and use of JAK inhibitors in treating AA. The basic pathomechanism of AA and the relevant role of JAK inhibitors should support how to efficiently use JAK inhibitors. We sought to investigate the effect of JAK1/2 inhibitor on an in vitro model of AA and to examine the possible mechanisms. We used interferon gamma-pretreated human dermal papilla cells (hDPCs) as an in vitro model of AA. Ruxolitinib was administered to the hDPCs, and cell viability was assessed. The change of expression of the Wnt/β-catenin pathway, molecules related to the JAK-STAT pathway, and growth factors in ruxolitinib-treated hDPCs was also examined by reverse transcription PCR and Western blot assay. We examined immune-privilege-related molecules by immunohistochemistry in hair-follicle culture models. Ruxolitinib did not affect the cell viability of the hDPCs. Ruxolitinib activated several molecules in the Wnt/β-catenin signaling pathway, including *Lef*1 and *β-catenin*, and suppressed the transcription of *DKK1* in hDPCs, but not its translation. Ruxolitinib reverted IFN-γ-induced expression of *caspase-1*, *IL-1β*, *IL-15*, and *IL-18*, and stimulated several growth factors, such as *FGF7*. Ruxolitinib suppressed the phosphorylation of JAK1, JAK2 and JAK3, and STAT1 and 3 compared to IFN-γ pretreated hDPCs. Ruxolitinib pretreatment showed a protective effect on IFN-γ-induced expression of MHC-class II molecules in cultured hair follicles. In conclusion, ruxolitinib modulated and reverted the interferon-induced inflammatory changes by blocking the JAK-STAT pathway in hDPCs under an AA-like environment. Ruxolitinib directly stimulated anagen-re-entry signals in hDPCs by affecting the Wnt/β-catenin pathway and promoting growth factors in hDPCs. Ruxolitinib treatment prevented IFN-γ-induced collapse of hair-follicle immune privilege.

## 1. Introduction

Alopecia areata (AA) is understood as an autoimmune hair-loss disorder that results from the collapse of hair-follicle immune privilege [1,2]. So far, the pathomechanism of chronic AA and alopecia totalis or alopecia universalis is not completely understood. Most patients with alopecia totalis or alopecia universalis are refractory to the conventional treatment.

The Janus kinase (JAK) signal transducers and activators of transcription protein (STAT) pathway and its signal transduction downstream cytokines, such as interleukin (IL)-15 and interferon (IFN)-**γ**, are considered to be key players in the autoimmune inflammatory mechanism in AA [3]. CD8+ cytotoxic T cells expressing natural killer group 2D pathogenic T cell proliferation around the HF was proven by activating JAK/STAT signaling in AA [4]. The effects of the JAK inhibitor in AA have been considered to result from an anti-inflammatory effect and immune-modulation on pathogenic T cells, which surround the hair follicles [5]. However, whether chronic AA patients with sparse peribulbar T cells could be indicated to JAK inhibitors should be determined by examining the effects of JAK inhibitors on other hair cells, such as dermal papilla, hair germ, and bulge.

Several kinds of JAK inhibitors that are currently available target JAK1/2, JAK1/3, and tyrosine kinase. JAK 1/3 and 1/2 inhibitors are US Food and Drug Administration approved for use in treating rheumatoid arthritis, myelofibrosis, and polycythemia vera [6,7,8]. Tofacitinib is a representative pan JAK inhibitor and strongly blocks JAK1/3 but weakly inhibits JAK2. Ruxolitinib inhibits JAK 1 and 2 to a similar extent but inhibits JAK 3 only slightly. The use of JAK inhibitors in AA is currently off-label, and worldwide clinical trials with JAK1/2 and JAK1/3 inhibitors to treat AA are under investigation [9,10,11,12]. After taking JAK inhibitors, some AA patients show dramatic hair regrowth; however, not every patient responds to the treatment, and a significant portion of the patients experience recurrence after they discontinue the treatment with JAK inhibitors [9,10,11,12]. Patients that are expected to have good responses and should be given JAK inhibitors should be further studied. More clinical and experimental evidence is needed to establish a safe and effective treatment protocol, such as duration, dosage, and administration route to induce remission and maintenance.

AA is considered to be a hair-cycling disorder that has some defects in anagen re-entry. Restoring the hair-follicle immune privilege upregulates genes responsible for cell proliferation and differentiation in hair cells and induction of anagen re-entry. The Wnt/β-catenin signaling pathway plays key roles in hair cycling [13]. So far, only a few studies have reported the role of the Wnt/β-catenin signaling pathway in hDPCs in response to JAK inhibitors [14].

We investigated the effect of a JAK1/2 inhibitor, ruxolitinib, on the viability of human dermal papilla cells (hDPCs) with or without pretreatment by interferon (IFN). We focused on the activation of JAK/STAT and the Wnt/β-catenin signaling pathway, the change of expression of hair-follicle immune privilege and several cytokines and growth factors by ruxolitinib.

## 2. Results

### 2.1. Effects of Ruxolitinib on the Cell Viability of hDPCs Treated with IFN-γ

We investigated whether ruxolitinib increased the viability of hDPCs. In order to identify a dose dependent effect, we treated hDPCs with 1 to 10,000 nM of ruxolitinib. Cell viability was assessed by MTT assays, as shown in Figure 1. Whereas IFN-γ significantly downregulated cell viability of hDPCs, ruxolitinib alone or with IFN-γ at various concentrations did not significantly affect cell viability (Figure 1). A total of 100 nM of ruxolitinib was used in overall experiments in this study.

### 2.2. Effects of Ruxolitinib on the mRNA Expression of Genes Related to Anagen Re-Entry or Anagen Arrest in IFN-γ-Treated hDPCs

Next, we investigated the effect of exposure to ruxolitinib on IFN-γ-related genes at mRNA levels, which included *caspase-1*, *IL-1β*, *IL-15* and *IL-18*. IFN-γ significantly upregulated *caspase-1*, *IL-1β*, *IL-15* and *IL-18* genes, and ruxolitinib effectively reverted these changes (Figure 2).

### 2.3. Effects of Ruxolitinib on the Protein Expression of JAK/STAT Pathway-Related Molecules in IFN-γ-Treated hDPCs

We examined the change of the JAK/STAT signaling pathway in hDPCs; it is the correct inhibitory target of ruxolitinib. Since ruxolitinib is known to be a JAK1/2 inhibitor, it completely blocked phosphorylation of JAK1 and 2, but it is also known to partially inhibit JAK3 phosphorylation in hDPCs. The levels of phosphorylation of STAT1 was partially suppressed by ruxolitinib. Overall, IFN-γ-induced expression of JAK1, 2, and 3, and STAT1, 3, and 5 proteins was significantly increased compared to control and suppressed by ruxolitinib at phosphorylated and sometimes total protein levels in the western blots. The inhibitory effects of ruxolitinib on JAK1, 2 and 3 were statistically significant compared to IFN 2 h-treated control. The expression of STAT1, 3, and 5 was declined by ruxolitinib. However, the inhibitory effect of ruxolitinib on the expression of STAT1, 3, and 5 was not significant compared to IFN 2 h-treated control.

The levels of phosphorylation of DKK1 were increased by short-term treatment with IFN-**γ** and normalized by ruxolitinib, but not significantly (Figure 3). We expected the levels of phosphorylation of glycogen synthase kinase (GSK)-3β, β-catenin in hDPCs to increase after treatment with ruxolitinib; however, the change of expression was not significant, unlike their changes in mRNA levels.

### 2.4. Effects of Ruxolitinib on the mRNA Expression of Wnt/β-Catenin Signaling Pathways in IFN-γ-Treated hDPCs

Genes responsible for β-catenin/Wnt signaling including *Lef1*, *β-catenin, Wnt7*, *TGF-β2* and *DKK1* mRNA were investigated by means of RT-PCR. We found a significant increase of mRNA of *Lef1* and *β-catenin* and a decrease of *DKK1* and *TGF-β2* two hours after treatment with ruxolitinib in hDPCs.

Treatment with ruxolitinib alone did not affect the levels of *Wnt7*, *Lef1*, and *DKK1* mRNA compared to control; however, the effect of ruxolitinib on IFN-γ pretreated hDPCs significantly reverted the effect of IFN-γ. Ruxolitinib treatment alone significantly increased the expression of *β-catenin* mRNA compared to control. However, the suppression was not significantly restored by addition of ruxolitinib treatment, in hDPCs (Figure 4). However, the increased transcription of *β-catenin* mRNA did not lead to increased translation of β-catenin (Figure 3).

Treatment with ruxolitinib alone significantly suppressed the mRNA expression of *TGF-β* in hDPCs, whereas IFN-γ treatment significantly enhanced *TGF-β* mRNA expression compared to the control, and ruxolitinib reverted the IFN-γ-induced *TGF-β* gene expression (Figure 4).

### 2.5. Effects of Ruxolitinib on the mRNA Expression of Growth Factors in IFN-γ-Treated hDPCs

We examined the change of FGF2, FGF7, IGF1, PDGF, and VEGF mRNA expression by ruxolitinib in hDPCs. All the growth factors analyzed, except FGF7, were significantly suppressed by IFN at the mRNA level and returned to control values by ruxolitinib. Interestingly, ruxolitinib treatment alone or together with IFN-γ significantly stimulated FGF7 expression (Figure 5).

### 2.6. Effects of Ruxolitinib on the Protein Expression of HF-IP-Related Molecules in IFN-γ-Treated hDPCs (IF) in a Mouse Model of Vibrissa HF Organ Culture

IFN-**γ** treatment significantly induced the expression of MHC class II molecules in the mouse vibrissa hair bulbs and the surrounding follicular sheath as compared with the controls. In protection assay, when ruxolitinib was given before treatment with IFN-γ, the expression of the MHC class II molecule was significantly downregulated in the hair-follicle organ culture compared to IFN-γ only treated group. However, in the restoration assay, ruxolitinib administered after treatment with IFN-γ was not enough to inhibit the effect of IFN-γ on the MHC class II molecule induction in mouse HFs (Figure 6).

## 3. Discussion

We intended to investigate how ruxolitinib could affect the new hair cycle onset and hair growth-promoting microenvironment in IFN-γ-treated hDPCs, similar to the AA microenvironment. Our results demonstrate that ruxolitinib did not diminish hDPC viability. Treatment with ruxolitinib alone did not stimulate anagen-re-entry-related molecules such as *Wnt7*, *Lef1* and *DKK1*, but effectively reverted IFN-γ-induced changes in the transcription of Wnt/β-catenin in hDPCs. Ruxolitinib effectively prevented HF-IP collapse by IFN-γ treatment and partially recovered the HF-IP in a murine hair-follicle organ-culture model.

Hair bulb and bulge interaction is important for starting a new anagen cycle. DPCs play a key role in the interaction to induce initiation of the anagen phase of the hair cycle [13]. The Wnt/β-catenin signaling pathway is the major pathway in activation of hDPCs to enter into the anagen re-entry. β-catenin is required as a proliferation signal in bulge stem cells. Interestingly, treatment with ruxolitinib alone significantly stimulated the expression of *β-catenin* mRNA beyond the level of reversing the down-regulation of β-catenin mRNA induced by IFN. However, it did not lead to translation of β-catenin, which suggests a more complex mechanism of post-transcriptional regulation of β-catenin. JAK/STAT signaling inactivation in hDPCs seems to induce expression of many growth factors and suppress some proinflammatory cytokines, which leads to a hair-growth-promoting microenvironment.

It is known that JAK1/2 signaling activates the phosphorylation of STAT 1/3 and JAK1/3 signaling that are more involved in the phosphorylation of STAT1/5, which subsequently leads to inflammatory cytokine secretion in T cells. Although the major source of inflammatory cytokines is pathogenic T cells, not hDPCs, IFN-γ-induced *caspase-1*, *IL-1β*, *IL-15* and *IL-18* mRNA, signature proinflammatory cytokines of AA were suppressed by ruxolitinib treatment in hDPCs.

The role of JAK/STAT signaling in the hDPCs is not well known. JAK1/2 signaling is suggested as being related to cell survival and stress response in hDPCs [15]. The role of JAK1/3 signaling in hDPCs is controversial. Legrand et al. reported that STAT5 plays an important role in inducing the hair-growth phase in DPCs, and the activity of STAT5 is not affected by inhibition of JAK1/2 [16]. Harel et al. demonstrated that JAK2, STAT3, and STAT5 are all upregulated in the DP as compared to the bulge and the hair germ in mouse models, that activated STAT5 in DP could be related to catagen induction, and that phosphorylated STAT3 is detected in hair germ cells during the catagen and telogen period [14]. IFN pretreatment on hDPCs can simulate an AA-like microenvironment. Our study found that the JAK/STAT signaling pathway is activated in response to IFN-treatment in hDPCs and ruxolitinib effectively reverts the phosphorylation of JAK1, 2 and 3 compared to IFN treated groups in hDPCs. Although overall downregulation of STAT1, 3, and 5 by ruxolitinib was not significant compared to IFN treated groups, the changes were identified in western blotting analysis.

JAK1/2 inhibitors are known to have weak inhibitory effects on JAK3, but we found a significant blockade of the phosphorylation of JAK3-STAT1/5 by ruxolitinib in hDPCs, which subsequently offset the proinflammatory effect by IFN. One may speculate that ruxolitinib-induced STAT5 inactivation may be a potential mechanism of the onset of new hair cycling by reversing catagen induction [14]. TGF-β is a well-known catagen inducer. In line with that, *TGF-β2* mRNA in hDPCs was significantly increased in response to IFN treatment and was diminished by ruxolitinib in this study.

Harel et al. [14] reported that tofacitinib promoted hair-growth rate when it was administered during the mid-telogen stage rather than the early telogen stage in mice. For the mechanism, they found that tofacitinib increased the expression of *TGF-β2*, *BMP6*, and *Lef1* in human DP spheres. Upregulated *TGF-β2* in the DP attenuates BMP signaling in the quiescence/activation step and contributes to early anagen initiation [14,17]. Interestingly, the inductivity of human DP spheres was improved by tofacitinib, but not by ruxolitinib, even though ruxolitinib increased the hair-growth rate in the organ culture model [14]. This suggests that ruxolitinib may prolong hair growth by positively affecting the survival of hDPCs, although we could not find that ruxolitinib-treated hDPCs improve cell viability. However, we found the activation of the Wnt/β-catenin signaling pathway including *Lef* and *β-catenin* and suppression of *DKK1* by ruxolitinib. Our results are consistent with previous studies, in that inhibition of JAK signaling is found to stimulate anagen re-entry signaling, but the effects on the expression of *TGF-β* was different from that of a previous study. This discrepancy may result from the cell-culture design, such as 2D culture and 3D spheroid culture methods, and the in vivo effect of ruxolitinib should be further studied.

We found the significant increase of *FGF7* mRNA expression by ruxolitinib in hDPCs. *FGF2*, *FGF7*, *IGF7*, *PDGF*, and *VEGF* mRNA expression was downregulated by IFN, and ruxolitinib has a growth-factor restorative effect, which was all suppressed by IFN-γ. FGF7 upregulation in DPCs is known to modulate BMP signaling, contribute to anagen initiation, and prolong the anagen period [14,18].

We proved that MHC class II molecule expression is induced in hair bulbs and around the hair follicles after exposure to IFN and is downregulated by pretreatment with ruxolitinib in an HF organ-culture model. Comparing the efficacy of the protective and restorative effects of HF-IP, the protective effect seems to be stronger than the restorative effect. These findings suggest that maintenance treatment of JAK inhibitors for severe AA patients may be helpful.

Our results demonstrated that ruxolitinib could stimulate anagen-re-entry-related molecules in hDPCs. Various growth factors, which could be involved in driving into anagen re-entry were increased in response to ruxolitinib. Ruxolitinib on hDPCs promotes recovery of the HF immune privilege in a hair-follicle organ-culture model of AA. Ruxolitinib potentially activates hDPCs and could be an attractive therapeutic option for patients with refractory AA at various stages.

## 4. Materials and Methods

### 4.1. Cell Culture

We purchased hDPCs from Promocell (GmbH, Heidelberg, Germany) and cultured them as previously described [1]. The hDPCs were seeded at a density of 2 × 10^5^ cells per well on 6-well culture plates in Dulbecco Modified Eagle Medium (DMEM, Gibco BRL, Life Technology, Karlsruhe, Germany) supplemented with 10% fetal bovine serum (FBS; Gibco BRL, Life Technology, Karlsruhe, Germany) and 1% penicillin/streptomycin (Gibco, BRL) for 24 h. The hDPCs in passages 3 to 4 were used for the experiments.

### 4.2. Cell Viability Assay

We performed a 3-(4,5-dimethylthiazol-2-yl)-2,5-diphenyltetrazolium bromide (MTT) assay to assess cell viability modifying the procedure described by Wasserman et al. [19]. Briefly, hDPCs (2 × 10^4^ cells/well) were seeded into 24-well plates. After 48 h of incubation with IFN-γ (100 ng/mL) or various concentrations of ruxolitinib, an MTT reagent was added and reacted for 4 h. Then, the samples were taken to measure absorbance at 540 nm with the enzyme-linked immunosorbent assay plate reader.

### 4.3. Real-Time PCR

Total RNA was extracted from the hDPCs using the TRIzol reagent (Invitrogen, Carlsbad, CA, USA) followed by cDNA synthesis with QuantiTect Rev. Transcription kit (Qiagen, Hilden, Germany) according to the manufacturer’s instructions. The cDNA was used for real-time Polymerase Chain Reaction (RT-PCR) using SYBR Green (Takara, Shiga, Japan). The primers sequences and PCR conditions are listed in Appendix A. The quantification of the PCR product was measured by using analysis software (Quantity one 1-D analysis, Bio-Rad, Hercules, CA, USA).

### 4.4. Western Blot Analysis

Cells were collected and lysed with RIPA lysis buffer (Pierce, Rockford, IL, USA). The amount of protein was measured with Bradford reagent (Bio-Rad, Hercules, CA, USA) and compared with bovine serum albumin as a standard. Cell lysates containing the same amount of total protein were separated by electrophoresis on SDS-polyacrylamide gel and then transferred to a PVDF membrane. The membranes were blocked with 5% bovine serum albumin in TBS-T and subsequently incubated with primary antibodies. Primary antibodies against phospho-STAT 1,3,5, total-STAT 1,3,5, phospho-JAK1,2,3, total-JAK1,2,3, IL-15, phospho-GSK3 β, β-catenin, GAPDH and β-actin were purchased from Cell signaling Technology, Inc. (Cell Signal, Beverly, MA, USA) and Santa Cruz Biotechnology (Santa Cruz, CA, USA). Specific reactive bands were detected with a horseradish peroxidase-conjugated secondary antibody (Santa Cruz, CA, USA), and the immunoreactive bands were visualized by the enhanced chemiluminescence detection system (Millipore, Bedford, MA, USA).

### 4.5. Hair Follicle (HF) Organ Culture Assay

Six-week-old male C57BL/6 mouse vibrissae follicles were microdissected under sterile conditions, as previously described [20]. Individual follicles were placed in a 24-well tissue culture plate in Williams E medium, which was supplemented with hydrocortisone, insulin, and glutamine in the presence of DMSO, IFN-γ, or ruxolitinib. The culture medium was changed every two days, and images of individual follicles were filmed before and after ruxolitinib treatment. All procedures were carried out in accordance with the Institutional Animal Care and Use Committee—approved protocols (CUMC-2019-0061-02, date of approval: 24 July 2019).

### 4.6. Hair-Follicle Immune Privilege (HF-IP) Restoration/Protection Assay

In normal scalp-hair follicles, the expression of major histocompatibility complex (MHC) class I and II is detected only under disruption of HF-IP. Individual anagen VI HFs were microdissected and separated from mice snout skin and were cultured as an HF organ [21]. Two different HF organ cultural settings were done and compared somewhat as described by Kinori et al. [21]: in a “restoration assay”, HFs were pretreated with IFN-γ to induce IP collapse, and then ruxolitinib was added. Alternatively, in a “protection assay”, HFs were first treated with ruxolitinib, followed by the addition of IFN-γ. The effect of ruxolitinib on the expression of MHC class I and II (Genetex, INC., Irvine, CA, USA) in HFs was evaluated by immunohistochemistry. The immuno-positive cells were counted and converted to % format using the ImageJ analysis system (ImageJ; NIH, Bethesda, MD, USA).

### 4.7. Statistical Analysis

All data were expressed as mean ± SD. An ANOVA test and paired student’s *t*-test were used for statistical analysis. All tests were one-sided, and a *p* < 0.05 was considered statistically significant.

## Figures and Tables

**Figure 1 ijms-21-05137-f001:**
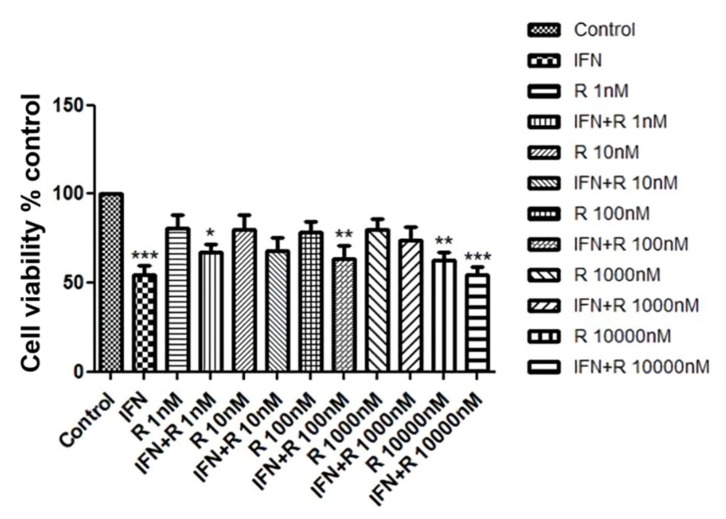
The effects of ruxolitinib treatment on the human dermal papilla cells (hDPCs) viability. Cell viability was not significantly affected by various concentrations of ruxolitinib treatment. The data represent the means ± SEM, *n* = 4, statistically significant at * *p* < 0.05, ** *p* < 0.01, *** *p* < 0.001, compared with the control.

**Figure 2 ijms-21-05137-f002:**
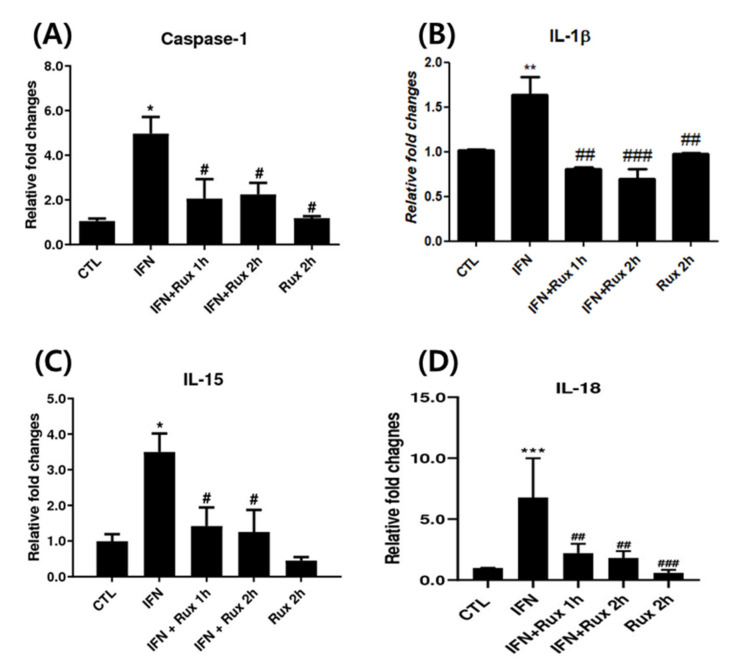
Ruxolitinib suppressed interferon (IFN)-γ-induced expressions of *caspase-1*, *interleukin (IL)-1 β*, and *IL-15*. The mRNA expression of (**A**) *caspase-1*, (**B**) *IL-1 β*, (**C**) *IL-15*, and (**D**) *IL-18* in hDPCs. The data represent the means ± SEM, *n* = 4, statistically significant at * *p* < 0.05, ** *p* < 0.01, *** *p* < 0.001 compared to control, and # *p* < 0.05, ## *p* < 0.01, ### *p* < 0.001 compared with IFN-γ-treated group.

**Figure 3 ijms-21-05137-f003:**
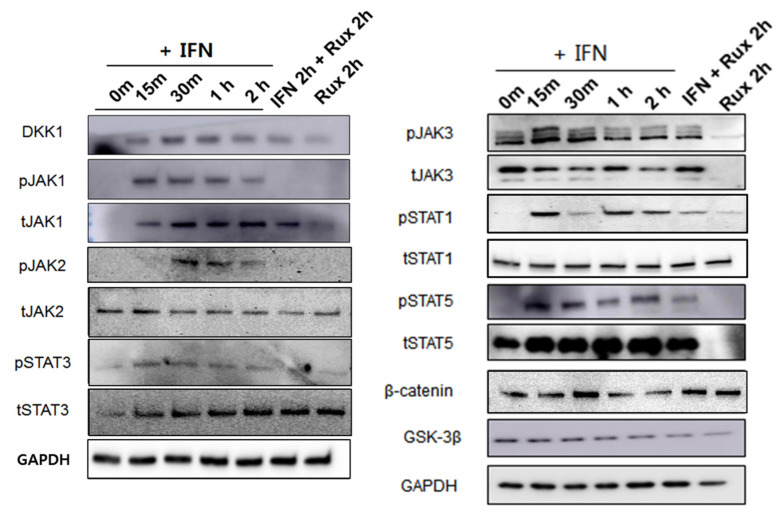
Changes in Janus kinase (JAK)-signal transducers and activators of transcription protein (STAT) pathway-related molecules after treatment with ruxolitinib. Western blotting showed changes in the levels of phosphorylated (**A**) JAK1, (**B**) JAK2, and (**C**) JAK3, (**D**) STAT1, (**E**) STAT3, and (**F**) STAT5, and (**G**) DKK1, (**H**) β-catenin, and (**I**) GSK-3β. The bands indicate serial protein expression levels up to 2 h after ruxolitinib treatment. The data represent the means ± SEM, *n* = 4, statistically significant at * *p* < 0.05, ** *p* < 0.01, *** *p* < 0.001 compared to control and # *p* < 0.05 compared with the IFN-γ-2 h treated group.

**Figure 4 ijms-21-05137-f004:**
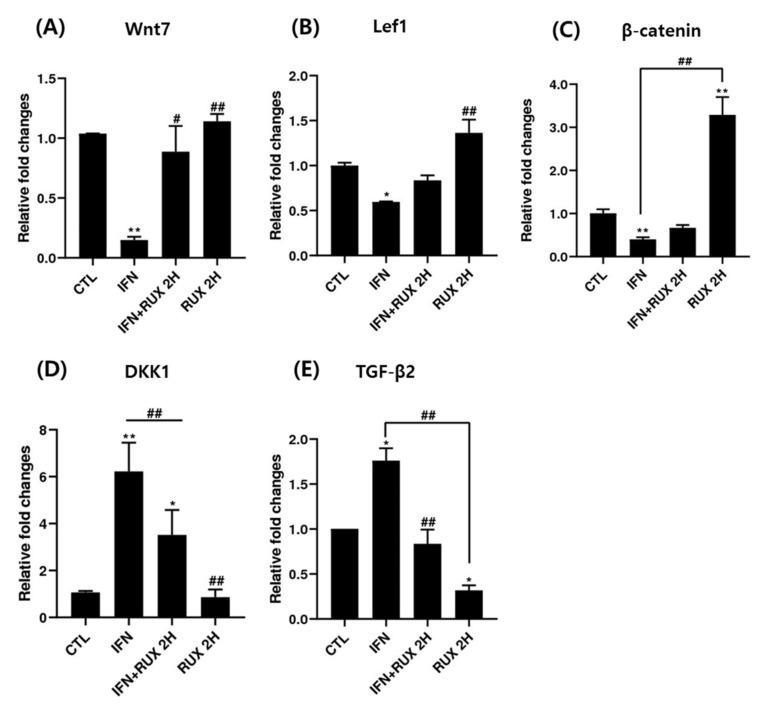
The effects of ruxolitinib treatment on molecules related to the Wnt/β-catenin signaling pathway. The mRNA expression of (**A**) *Wnt7*, (**B**) *Lef1*, (**C**) *β-catenin*, (**D**) *DKK1*, and (**E**) *TGF-β* in hDPCs. Ruxolitinib activated the mRNA expression of molecules related to the Wnt/β-catenin signaling pathway compared to the control and reverted the effect of IFN-γ in hDPCs. The data represent the means ± SEM, *n* = 3, statistically significant at * *p* < 0.05, ** *p* < 0.01 compared to the control and # *p* < 0.05, ## *p* < 0.01 compared to the IFN-γ-treated group.

**Figure 5 ijms-21-05137-f005:**
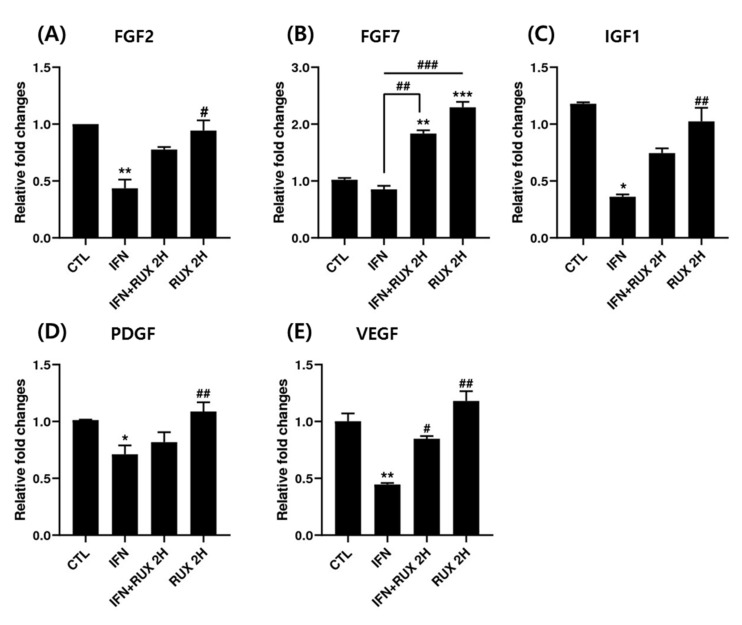
Effects of ruxolitinib treatment on hair-growth-related growth factors. Ruxolitinib significantly upregulated the mRNA of various growth factors including (**A**) FGF2, (**B**) FGF7, (**C**) IGF1, (**D**) PDGF, and (**E**) VEGF, which was suppressed by IFN-γ. The data represent the means ± SEM, *n* = 3, statistically significant at * *p* < 0.05, ** *p* < 0.01, *** *p* < 0.001, compared to the control and # *p* < 0.05, ## *p* < 0.01, ### *p* < 0.001 compared to the IFN-γ-treated group.

**Figure 6 ijms-21-05137-f006:**
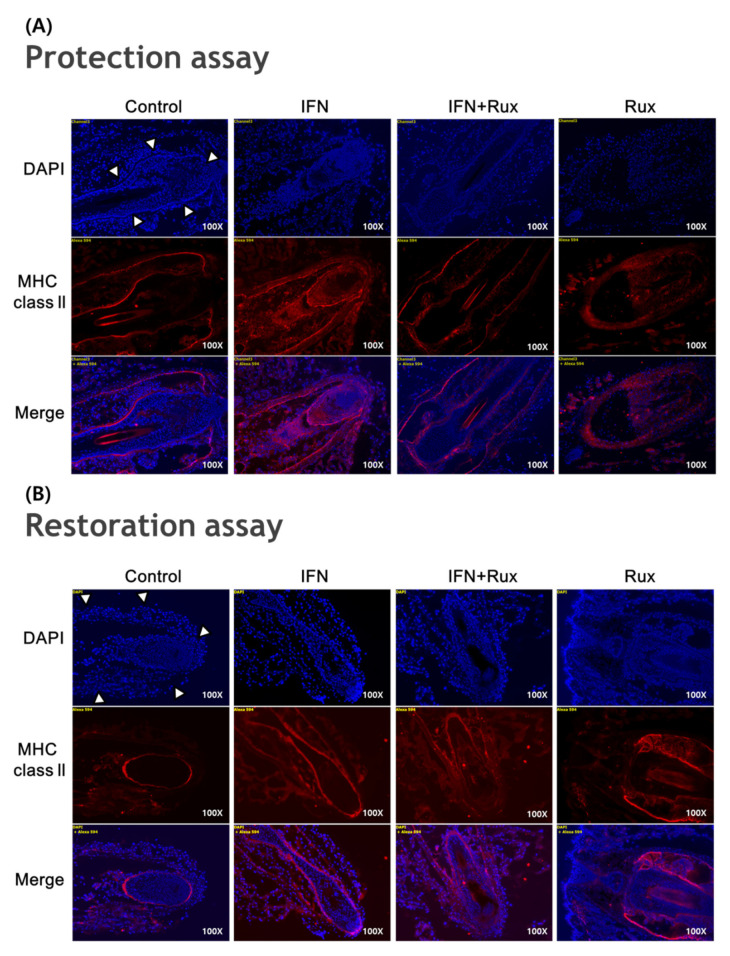
Effects of ruxolitinib treatment on hair-follicle organ culture. (**A**) Ruxolitinib pretreatment significantly prevented the induction of MHC class II molecule by IFN-γ in and around the hair follicles in the protection assay; (**B**) The expression of MHC class II molecule in the hair follicles ruxolitinib administered after treatment with IFN-γ was not effectively suppressed in the restoration assay; (**C**) Relative number of immune-positive cells of MHC class II molecule in the protection and restoration assay. White arrowheads indicate outline of hair follicles. The data represent the means ± SEM, *n* = 3, statistically significant at * *p* < 0.05, and ** *p* < 0.01 compared to the control and # *p* < 0.05 compared to the IFN-γ-treated group.

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
