# Peer review of "The Effect of JAK Inhibitor on the Survival, Anagen Re-Entry, and Hair Follicle Immune Privilege Restoration in Human Dermal Papilla Cells"

_ijms, 2020, doi:10.3390/ijms21145137_

Round 1
Reviewer 1 Report
This manuscript by Kim et al describe their efforts to characterize the JAK inhibitor ruxolitinib on human dermal papilla cells (hDPCs). They make a reasonable case for the idea that the drug might have an effect in these cells and could be useful in treating alopecia areata, an autoimmune disorder that has links to interferon gamma (IFN), which signals through the JAK/STAT pathway. They cultured hDPCs and examined effects of ruxolitinib treatment on viability and gene/protein expression, and they used mouse hair follicles in culture to look at immune effects. The results are interesting, and the revised manuscript has improved, resolving most of my prior concerns. However, some different statistical analysis and comparisons are needed to interpret the data correctly, as detailed below.
Figure 1 has been improved by more samples and corrected labeling, although this remains hard to interpret. It would be more informative to show statistical comparisons between similar samples, not just the control, since it is unclear if there is a difference with or without the IFN treatments. This could be done by additional pairwise comparisons or an ANOVA test on the whole set. It is possible it was already done in pairwise comparisons but I was not sure since it said its compared to control, which seems to be the first column.
In figure 3, the western blots seem to show decreases in pJAK1, pJAK2 and pSTAT1 and pSTAT5 with ruxolitinib treatment of IFN cells compared to 2hr IFN alone, and possibly a decrease in total JAK2 and total STAT5 and in increase in total JAK3. However, the total protein levels aren’t shown quantitatively, which makes the data a bit hard to interpret. The authors claim in the text that they can’t make conclusions about cases in which the total protein was lost with ruxolitinib treatment, but this does not matter as those proteins are also not there without IFN treatment. So the important comparisons should be 2hr IFN alone compared to IFN+ ruxolitinib treatment, and in those cases the relevant proteins are present. This existing data needs to be presented.
According to the graphs, only pJAK3/tJAK3 is significantly different with ruxolitinib+IFN compared to IFN alone (with the # for significance), although this likely has to do with the change in total protein, not the phosphorylation/activity. It seems likely from the graphs that pJAK1/tJAK1 and pJAK2/tJAK2 are also significantly suppressed by ruxolitinib but since the statistics aren’t presented it is not clear. It does not look like there is significant suppression of any of the phosphoSTATs with ruxolitinib. Thus, the claims in lines 95-99 need to be revised to reflect the data better. Likewise line 204 of the discussion should be revised.
In figures 4 and 5 the authors need to indicate the statistical significance (if there is any) of the difference between IFN treated and IFN+ ruxolitinib in all cases (including Wnt7, PDGF). Line 122 is hard to understand, but could be changed to something like “Ruxolitinib treatment alone significantly increased the expression of β-catenin mRNA. However, the suppression of this transcript by IFN-γ was significantly restored by addition of Ruxolitinib treatment, in hDPCs (Fig4).”
Details of figure 6 are improved but the images and labels are very small and would be clearer if enlarged.
Author Response
This manuscript by Kim et al describe their efforts to characterize the JAK inhibitor ruxolitinib on human dermal papilla cells (hDPCs). They make a reasonable case for the idea that the drug might have an effect in these cells and could be useful in treating alopecia areata, an autoimmune disorder that has links to interferon gamma (IFN), which signals through the JAK/STAT pathway. They cultured hDPCs and examined effects of ruxolitinib treatment on viability and gene/protein expression, and they used mouse hair follicles in culture to look at immune effects. The results are interesting, and the revised manuscript has improved, resolving most of my prior concerns. However, some different statistical analysis and comparisons are needed to interpret the data correctly, as detailed below.
A) Thank you for your valuable comments. Our article was revised focusing on the correct statistical analysis and comparisons according to reviewers’ comments. We revised a few suggested figures and results (Fig.3, Fig.4, Fig.5, and Fig.6). We are convinced that the revised article is substantially improved. Thank you, again.
Figure 1 has been improved by more samples and corrected labeling, although this remains hard to interpret. It would be more informative to show statistical comparisons between similar samples, not just the control, since it is unclear if there is a difference with or without the IFN treatments. This could be done by additional pairwise comparisons or an ANOVA test on the whole set. It is possible it was already done in pairwise comparisons but I was not sure since it said its compared to control, which seems to be the first column.
A)Thank you for the comments. That’s a very good point. All comparisons were done compared to control, which is the first column and * was marked if there is a statistical difference. Following your opinion, we additionally did pairwise comparisons and ANOVA test. However, it was proven that ruxolitnib did not significantly improve IFN-induced suppression of cell viability. To avoid confusion, the original Fig.1 was used as it was.
In figure 3, the western blots seem to show decreases in pJAK1, pJAK2 and pSTAT1 and pSTAT5 with ruxolitinib treatment of IFN cells compared to 2hr IFN alone, and possibly a decrease in total JAK2 and total STAT5 and in increase in total JAK3. However, the total protein levels aren’t shown quantitatively, which makes the data a bit hard to interpret. The authors claim in the text that they can’t make conclusions about cases in which the total protein was lost with ruxolitinib treatment, but this does not matter as those proteins are also not there without IFN treatment. So the important comparisons should be 2hr IFN alone compared to IFN+ ruxolitinib treatment, and in those cases the relevant proteins are present. This existing data needs to be presented.
According to the graphs, only pJAK3/tJAK3 is significantly different with ruxolitinib+IFN compared to IFN alone (with the # for significance), although this likely has to do with the change in total protein, not the phosphorylation/activity. It seems likely from the graphs that pJAK1/tJAK1 and pJAK2/tJAK2 are also significantly suppressed by ruxolitinib but since the statistics aren’t presented it is not clear. It does not look like there is significant suppression of any of the phosphoSTATs with ruxolitinib. Thus, the claims in lines 95-99 need to be revised to reflect the data better. Likewise line 204 of the discussion should be revised.
A) Thank you for your valuable comments. We conducted statistical processing twice in all the experiments. Basically, comparisons were made with control groups and additionally comparisons with IFN2h groups were done. The statistically significant value compared with control was *, and # compared to the IFN2h groups in the entire manuscript. Some # marks were found missing in Fig.3 which is the results of new experiments and revised. We apologize for the error. There was an omission in the process of drawing the significance mark # directly in the Figure due to basic statistical analyses are set to compare all groups to control.
Finally, the decrease in the phospho-/total protein ratio of JAK1, JAK2, and JAK3 with ruxolitinib treatment with or without IFN cells compared to 2hr IFN alone were found all significant. However, the same comparisons regarding STAT1, 3, and 5 were not significant when compared to 2hr IFN alone group. We revised the relevant sentences in the results (Line 95-99) and discussion section (Line 204) accordingly. Thank you again.
In figures 4 and 5 the authors need to indicate the statistical significance (if there is any) of the difference between IFN treated and IFN+ ruxolitinib in all cases (including Wnt7, PDGF). Line 122 is hard to understand, but could be changed to something like “Ruxolitinib treatment alone significantly increased the expression of β-catenin mRNA. However, the suppression of this transcript by IFN-γ was significantly restored by addition of Ruxolitinib treatment, in hDPCs (Fig4).”
A) Thank you for the comments. We revised Fig 4 and 5 to clearly indicate statistical significance of the difference between IFN treated and IFN+ ruxolitinib in all cases (including Wnt7, PDGF).
A) We appreciate your suggestion. Accordingly, we have revised the sentences in Line 122 as below: “Ruxolitinib treatment alone significantly increased the expression of β-catenin mRNA. However, the suppression of this transcript by IFN-γ was not significantly restored by addition of Ruxolitinib treatment, in hDPCs (Fig4).”
Details of figure 6 are improved but the images and labels are very small and would be clearer if enlarged.
A) Thank you. We have revised the images and labels accordingly.
Reviewer 2 Report
The revised manuscript has been improved.
I would suggest to wrap up the Abstract with a conclusive sentence instead of a descriptive result.
Author Response
The revised manuscript has been improved.
I would suggest to wrap up the Abstract with a conclusive sentence instead of a descriptive result.
A) Thank you for the valuable comments. We revised the conclusion of abstract more conclusive. “In conclusion, ruxolitinib modulated and reverted the interferon-induced inflammatory changes by blocking the JAK-STAT pathway in hDPCs under an AA-like environment. Ruxolitinib directly stimulated anagen-reentry signals in hDPCs by affecting the Wnt/β-catenin pathway and promoting growth factors in hDPCs. Ruxolitinib treatment prevented IFN-γ-induced collapse of hair-follicle immune privilege.”
This manuscript is a resubmission of an earlier submission. The following is a list of the peer review reports and author responses from that submission.
Round 1
Reviewer 1 Report
This manuscript by Kim et al describe their efforts to characterize the JAK inhibitor ruxolitinib on human dermal papilla cells (hDPCs). They make a reasonable case for the idea that the drug might have an effect in these cells and could be useful in treating alopecia areata, an autoimmune disorder that has links to interferon gamma, which signals through the JAK/STAT pathway. They cultured hDPCs and examined effects of ruxolitinib treatment on viability and gene/protein expression, and they used mouse hair follicles in culture to look at immune effects. While the results overall are interesting, the interpretation is limited because many of the results do not seem to rise to a high level of statistical significance and others seem confusing. So, some claims are not well supported by the data. The results do show a clear decrease in pJAK3, and pSTAT1 in response to the drug, and positive effects on a few Wnt signaling components and MHC class II-positive cells. It seems for multiple cases other that additional experiments or replicates are needed to verify and permit the conclusions.
Major concerns: the first data using an MTT assay looks at viability but the graph says % proliferation – these are not the same and this needs to be resolved. Ideally both viability and proliferation would be assessed through separate assays. Additionally, it is not clear there is a dose-dependent effect as is claimed. They provide an n=2 value here, which suggest the experiment was repeated twice but the differences are small, implying that more replicates could improve the confidence values.
In figure 2, the authors show a pretty strong decline in IL-18 RNA levels in response to IFNgamma, but they claim this gene was not affected (but that other targets are). If the variation is so large for IL-18 that this difference is not significant, It seems like more replicates are needed.
In terms of JAK/STAT signaling, they examine phosphorylated protein levels, and several seem to look disrupted by the Western data, but only pJAK3 and pSTAT1 are significantly different from the controls according to the quantifications. pJAK is missing a significance notation, but it looks to be quite different between IFN treated and IFN treated plus ruxolitinib. These results are described in the text and abstract in a misleading way to imply more components of the signaling differ significantly. Again, I would suggest this be repeated since, if the differences are small, more replicates might determine if the trends are due to the treatments. In fact for many targets 2 hours is a bit long to wait, it would be helpful to see the ruxolitinib treatment at an earlier timepoint. Also for many examples the total protein seems to drop a lot, which also impacts the relative phospho-levels and should be discussed.
They also claim that beta-catenin changes. Although the RNA is changed, the total protein is not (figure 3) so it is not clear this is relevant, or they have not identified the point in time when one reflects the other. Given the post-translational regulation of beta-catenin, the protein expression is very important.
It is not clear to me what the changes in growth factor RNA expression means- these seem to behave in all different ways in response to the different conditions and likely would be time-dependent in a way that was not explored. Additionally the protein levels may differ. to control levels with the drug, except FGF7, which increases. They are all significant but seem to have small changes, and it is not clear which might be most relevant to this system. Most of the RNA levels return close
Not enough detail is provided on how the MHC-class II immunohistochemistry was done to be confident that there are significant changes with the treatment; these results should be standardized and quantified. No results are shown here for class I so there should not be conclusions about its response to the IFNgamma signaling or the drug.
Reviewer 2 Report
I really appreciate this work
Reviewer 3 Report
This study is focused on the role of the JAK inhibitor ruxolitinib in the prevention and restoration of the effects caused by INF-γ, in an attempt to mimic AA in vitro and ex vivo and investigate the molecular mechanisms. The restoration of the mRNA expression levels of molecules related to the Wnt pathway as well as the induction of growth factors in response to the treatment with the inhibitor is well shown by the authors. However, the analysis of the expression of proteins related to the JAK/STAT pathway is one of the main weaknesses of the paper, as the results are not very conclusive. Finally, the prevention and restoration of the immune privilege ex vivo assays need to be better quantified in order to support the conclusions.
Fig. 1:
- Viability and proliferation is not the same. MTT assay reports cell viability. Label the graph axis according to the results presented. It should be also corrected in the discussion, unless proliferation assays are provided.
- Is the concentration of IFN constant in the assay?
- At ruxolitinib concentrations of 1-1000 nM, is the drop in cell viability not significantly different from the control?
Fig. 2:
- Why do the authors state in line 139 “ruxolitinib did not affect IL-18” but they show in fig 2 # indicating p-val < 0.05 when comparing IFN+Rux 2h with IFN-γ-treated group?
- What is the concentration of ruxolitinib used in this assay?
Fig. 3:
- Lines 149-150: “ruxolitinib (…) completely blocked phosphorylation of JAK1 and 2“. Fig. 3A, B: No significant differences are indicated in the graph. In the case of JAK2 phosphorylation, not even a trend. Also, even the opposite trend is shown in densitometry in B comparing P-JAK2/t-JAK2 in Rux 2h vs. Control. Please review these results and avoid overstatements
- Lines 151-153: “The levels of phosphorylation of STAT1 was partially suppressed by ruxolitinib. Interestingly, the levels of phosphorylation of STAT3 and 5 in hDPCs were completely inhibited by ruxolitinib”. This is very ambiguous. How do the authors explain that there is a residual expression of the phosphorylated form of STAT1 after the treatment with ruxolitinib, but total STAT1 expression is not detected? Likewise, STAT5 seems to be undergoing a dramatic decrease in expression in the presence of ruxolitinib, instead of showing changes in the phosphorylation levels. In contrast, STAT3 showed differential levels of the phosphorylated form, maintaining the level of expression of tSTAT3.
- Lines 154-155: “The levels of phosphorylation of DKK1 were increased by short-term treatment with IFNg and normalized by ruxolitinib (Fig. 3).” Since no significant differences are shown, the authors can only claim that the levels of DKK1 tend to increase by short term treatment with IFNg.
- Fig. 4: Clarify the statistical comparisons in B
- Lines 171-172: “Treatment with ruxolitinib alone did not affect DKK1 and TGF-β2 mRNA levels compared to control”. Fig. 4 evidences a significant decrease in TGF-β2 mRNA levels compared to the control (*)
- Lines 173-174: “Ruxolitinib did not change Wnt7 expression at mRNA levels in hDPCs, whereas IFN-γ treatment significantly suppressed the mRNA expression of Wnt7 compared to the control.” Although ruxolitinib alone does not change the mRNA levels of Wnt7 compared to the control, ruxolitinib did indeed revert the decrease in Wnt7 mRNA levels observed as a consequence of the treatment with IFN-γ. Is there any reason not to mention this?
- It is confusing that specially in the images shown in the restoration assay, ruxolitinib alone seems to increase the expression of MHC II molecules. Therefore, it is unlikely that ruxolitinib will revert the effect of IFN-γ. How do the authors interpret this? How many follicles were included in the protection and restoration assays?
- The authors should provide quantifications that support the conclusions of the immunofluorescence assays.
Minor points:
- Italics for genes (Fig. 4)
- “Reversed” should be replaced with “reverted”
- Lines 196, 197, 202, 234: “mice” should be replaced with mouse